# Altered Posttranslational Modification of Microtubules Contributes to Disturbed Enterocyte Morphology in Celiac Disease

**DOI:** 10.3390/ijms24032635

**Published:** 2023-01-30

**Authors:** Sebastian Stricker, Manuel Müller, Klaus-Peter Zimmer, Ralf Jacob

**Affiliations:** 1Department of Pediatrics, Justus-Liebig-University Giessen, 35392 Giessen, Germany; 2Department of Cell Biology and Cell Pathology, Philipps-University Marburg, 35032 Marburg, Germany

**Keywords:** celiac disease, microtubules, detyrosinated tubulin, tubulin tyrosine ligase, vasohibin-2

## Abstract

Celiac disease (CD) represents a frequent autoimmune disease triggered by the ingestion of gliadin in genetically predisposed individuals. The alteration of enterocytes and brush border membrane morphology have been repetitively demonstrated, but the underlying mechanisms remain unclear. Microtubules represent a major element of the cytoskeleton and exert multiple functions depending on their tyrosination status. The aim of our study was to investigate whether posttranslational modification of microtubules was altered in the context of CD and whether this mechanism contributed to morphological changes of CD enterocytes. We examined the expression of tubulin tyrosine ligase (TTL) and vasohibin-2 (VASH2) and the level of detyrosinated and acetylated tubulin in duodenal biopsies and Caco-2 cells by immunoblot and immunofluorescence microcopy. Electron microscopy was performed to investigate the subcellular distribution of detyrosinated tubulin and brush border membrane architecture in CD biopsies and Madin–Darby Canine Kidney type II (MDCK) cells lacking TTL. CD enterocytes and Caco-2 cells stimulated with digested gliadin or IFN-y displayed a flattened cell morphology. This disturbed cellular architecture was accompanied by an increased amount of detyrosinated and acetylated tubulin and corresponding high expression of VASH2 and low expression of TTL. The altered posttranslational modification of tubulin was reversible after the introduction of the gluten-free diet. CD enterocytes and MDCK cells deficient in TTL displayed a reduced cell height along with an increased cell width and a reduced number of apical microvilli. Our results provide a functional explanation for the observed morphological alterations of the enterocytes observed in CD and provide diagnostic potential of the tyrosination status of microtubules as an early marker of villous atrophy and CD inflammation.

## 1. Introduction

Celiac disease (CD) is a chronic inflammatory, multisystemic disease triggered by the ingestion of wheat gliadin and related prolamins in genetically predisposed individuals carrying the HLA (human leukocyte antigen)-DQ2/8 genotype [1,2]. It represents one of the most frequent autoimmune diseases and affects about 1% of the Western civilization with increasing incidence in the last decades [3,4]. From a clinical perspective, CD appears with multifaceted intestinal and extraintestinal manifestations. Abdominal pain, bloating, and diarrhea represent classical intestinal symptoms that often dominate in toddlers, whereas extraintestinal symptoms such as amenorrhea, delayed puberty, autoimmune hepatitis, ataxia, and osteopenia are often observed in older patients [5].

On a molecular level, gliadin peptides are resistant to non-enzymatic and enzymatic digestion in the upper gastrointestinal tract and reach the intestinal lamina propria via paracellular and transcellular pathways. Immunogenic gliadin peptides are deamidated by transglutaminase 2, the recognized autoantigen of CD. This enzymatic modification markedly increases their immunogenicity and is a precondition for the subsequent antigen presentation via HLA-DQ2/8 receptors [6,7]. The presentation of immunogenic gliadin peptides initiates an inflammatory cascade characterized by the production of pro-inflammatory cytokines (e.g., IFN-y) and autoantibodies. In addition, non-immunogenic gliadin peptides activate the innate immune response and exert direct toxic effects on enterocytes [8].

The characteristic histopathological changes of the duodenal mucosa in CD include an increased amount of CD3+ intraepithelial lymphocytes, crypt hyperplasia, and villous atrophy [9]. However, changes of enterocyte morphology and alterations of microvilli architecture have been reported as well [10,11,12]. Currently, the molecular mechanisms involved in the morphological changes of the enterocytes in CD are not known, but microtubules, which represent one major element of the cytoskeleton, might play a relevant role. Even though their molecular structure consisting of α,β-tubulin heterodimers is highly conserved, microtubules can exert a vast range of diverse functions (cell shape, motility, intracellular transport, and cell division) [13,14]. Those functions are often regulated by posttranslational modification (PTM) of α-tubulin, including acetylation, detyrosination, and many others [14]. For instance, acetylation of Lys-40 and detyrosination at the carboxy-terminus of α-tubulin were associated with rather stable microtubules [15,16]. Microtubules rich in detyrosinated tubulin (also called detyr-tubulin) are more resistant to depolymerizing agents (e.g., nocodazole) [17], whereas microtubules containing high amounts of tyrosinated tubulin (tyr-tubulin) are more dynamic and show a turnover halftime of less than 10 min [18,19]. Tyrosination of the α-tubulin subunit is catalyzed by one specific enzyme called tubulin tyrosine ligase (TTL), whereas detyrosination is mediated by vasohibin (VASH) 1 and 2 and microtubule-associated tyrosine carboxypeptidase (MATCAP) [20,21]. The tyrosination status of α-tubulin plays a special role in neurodevelopment, which is shown by the fact that TTL-deficient mice die perinatally due to a massively impaired brain architecture [22]. Furthermore, detyrosination affects mitosis and was shown to be associated with aggressive subtypes of breast cancer [22,23].

Recently, we have shown that a complete knockdown of TTL results in an increased amount of intracellular detyr-tubulin along with a reduced cell height and increased cell width in Madin–Darby Canine Kidney type II (MDCK) cells [24]. Moreover, enterocytes of intestinal biopsies distributed along the crypt villus axis in the small intestine also exhibited TTL-dependent changes in cell shape [24]. We thus hypothesized that an altered PTM of microtubules might contribute to the disturbed architecture (i.e., flattening) of CD epithelium in the gut.

To address this issue, we investigated the tyrosination status of tubulins and cellular architecture in CD biopsies and stimulated Caco-2 cells.

## 2. Results

### 2.1. Increased Levels of Detyr-Tubulin and Disturbed Cellular Morphology in CD Epithelium

First, we addressed the question of whether the tyrosination status of microtubules differed between control patients and patients with active CD on a gluten-containing diet. For this purpose, we studied the expression of tyr- as well as detyr-tubulin by immunofluorescence on tissue sections obtained from duodenal biopsies. We found significantly enhanced quantities of detyr-tubulin in the mucosa of CD patients, whereas the amount of tyr-tubulin was reduced (Figure 1A,B). This observation was accompanied by morphological alterations of enterocytes. Enterocytes of the CD epithelium revealed a trend toward reduced cell height (30 ± 4 µm vs. 26 ± 3 µm; n.s.) and a significantly increased cell width (5.8 ± 0.5 µm vs. 8.0 ± 0.5 µm; *p* < 0.01; Figure 1C). Thus, CD enterocytes with active disease had a more flattened and stocky shape, which was reflected by a reduced aspect ratio (height/width) (Figure 1C).

To confirm this observation and to obtain a more detailed insight into the ultrastructure, we used the cryoimmunogold method to examine the subcellular localization of detyr-tubulin in ultrathin-frozen cryosections of the duodenal epithelium. The presence of detyr-tubulin was most prominent within the apical compartment of enterocytes in CD patients as well as in control patients (Figure 2A,B). In agreement with the immunofluorescence data, intracellular concentrations of detyr-tubulin were higher in enterocytes from CD compared with control patients (0.8 ± 0.3 vs. 0.4 ± 0.2 GP/µm^2^; *p* < 0.05; Figure 2C). These observations were accompanied by an altered microvilli architecture in CD patient biopsies. Here microvilli displayed a rather short and bulky shape, and their absolute numbers appeared to be reduced in comparison with controls (Figure 2A,B).

### 2.2. Differential Posttranslational Modification of Tubulin in CD Patients with Active Disease and in Remission

Next, we assessed the protein quantities of modified tubulins in biopsy samples of patients with active CD and CD patients in remission under a gluten-free diet by immunoblot. The quantities of tyr-tubulin were not affected by the disease status and did not differ between control patients and patients with CD irrespective of their diet (Figure 3A–C). In contrast, the amount of acetylated tubulin (acetyl-tubulin) was significantly increased in patients with untreated CD and dropped back to the level of control patients following administration of the gluten-free diet (Figure 3A–C). Moreover, the expression of tubulin tyrosine ligase (TTL), the enzyme catalyzing tubulin tyrosination, was significantly reduced in patients with active CD (Figure 3A–C). However, the expression of TTL was restored in CD patients after the introduction of the gluten-free diet to the level observed in control patients (Figure 3A–C). Inversely to the expression of TTL, the amount of detyr-tubulin significantly increased in active CD but dropped back to the level of control patients in CD patients following a gluten-free diet (Figure 3A–C). Thus, we displayed a significantly reduced expression of TTL and a higher amount of posttranslationally modified acetylated and detyrosinated tubulin in active CD. The fact that the observed changes were reversible when a gluten-free diet was administered further denotes that an altered PTM of microtubules might be a specific feature of CD inflammation.

### 2.3. Stimulation of Caco-2 Cells with Digested Gliadin and IFN-y Alters PTM of Microtubules and Affects Cellular Morphology

The observation of an altered tyrosination of α-tubulin accompanied by disturbed enterocyte morphology in CD epithelium prompted us to further elucidate the underlying mechanisms in a cell culture model. For this purpose, we chose the immortalized gastrointestinal epithelial cell line Caco-2, which adequately reflects the physiology of differentiated enterocytes of the small intestine [25]. We stimulated Caco-2 cells 7–10 days after confluency with different inflammatory agents. First, we analyzed the effect of digested gliadin (Frazer’s Fraction III (FF)) containing immunogenic and toxic peptides of wheat gliadin. Immunoblots of cell lysates demonstrated a dose-dependent increase in the amount of detyr-tubulin in Caco-2 cells at 100 µg/mL FF (1.3 ± 0.1; *p* < 0.05) and 1000 µg/mL FF (1.6 ± 0.2; *p* < 0.01) (Figure 4A,B). This effect was specific for FF since treatment with ovalbumin, which was used as a control antigen, did not affect the level of detyr-tubulin (Figure 4A,B). Similarly, the amount of acetyl-tubulin significantly increased after treatment with FF, whereas OVA did not affect acetyl-tubulin level (Figure 4A,B). FF treatment did not alter TTL expression. However, FF stimulation caused a significant increase in VASH2 expression, an enzyme known to promote detyrosination (7.8 ± 1.9, *p* < 0.01; Figure 4A,B). Expression of both enzymes was not affected by treatment with OVA, which underlined the CD specificity of our observations.

Next, we assessed the effect of digested gliadin on TTL expression and cell shape by immunofluorescence microscopy. The expression of TTL was significantly reduced after stimulation with FF. Correspondingly, the level of detyr-tubulin increased after FF treatment (Figure 4C,D). This alteration in the PTM of microtubules was accompanied by morphological changes of Caco-2 cells. Z-stack images of FF-stimulated Caco-2 cells revealed a significantly reduced cell height (20 ± 2 vs. 15 ± 1 µm; *p* < 0.001; Figure 4C,E). Additionally, they displayed an increased cell area (186 ± 40 vs. 300 ± 30 µm^2^; *p* < 0.01; Figure 4C,E). Hence, Caco-2 cells adopted a rather flat and bulky morphology after FF treatment, which is in line with our observations on biopsies from CD patients on a gluten-containing diet.

Then, we analyzed the effect of interferon-y (IFN-y) on the PTM of microtubules and the morphology of Caco-2 cells. IFN-y represents the key cytokine of Th1-mediated inflammation in CD. It was released by CD4+ T cells upon the presentation of immunogenic gliadin peptides and exerted multiple effects on the epithelial and immune cells of the duodenal mucosa. On the protein level, we observed a significant, dose-dependent increase in the cellular level of detyr-tubulin after 48 h of treatment with 500 IU/mL IFN-y (4.9 ± 1.9, *p* < 0.01; Figure 5A,B). Similarly, the cellular level of acetyl-tubulin was increased by treatment with 500 IU/mL IFN-y (4.0 ± 1.4, *p* < 0.05; Figure 5A,B). Expression of TTL was not significantly altered by treatment with IFN-y, whereas the VASH2 level increased in a dose-dependent manner (3.4 ± 1.4, *p* < 0.05; Figure 5A,B). On the microscopical level, we confirmed the increased levels of detyr-tubulin and a corresponding reduced expression of TTL after IFN-y stimulation (Figure 5C,E). Detyr-tubulin enriched microtubules were predominantly detected in subapical areas above the cell nuclei (Figure 5D). In addition, we observed an altered cell shape with reduced cell height (24 ± 4 vs 13 ± 3 µm, *p* < 0.001; Figure 5C; z-stack) and an increased cell area (191 ± 44 vs 285 ± 34 µm^2^, *p* < 0.05; Figure 5F) in treated Caco-2 cells. The altered tyrosination status as well as the flattened cell shape corresponded to the observations made in CD epithelium and Caco-2 cells after treatment with digested gliadin. In summary, we demonstrated that CD-associated inflammatory triggers (digested gliadin, IFN-y) but not unspecific alimentary antigens (ovalbumin) induced an altered PTM of α-tubulin resulting in morphological changes of Caco-2 cells.

### 2.4. Knockout of TTL Disturbs Brush Border Membrane Architecture in MDCK Cells

To further address the effect of the tyrosination status of microtubules on the architecture of the brush border membrane, we performed electron microscopical imaging of polarized wild-type MDCK cells and MDCK cells lacking TTL (MDCK_∆TTL_). The efficient knockout of TTL using CRISPR/Cas9 and the corresponding increase in detyr-tubulin levels have been extensively described in our previous publications [24,26].

Using epon-embedded ultrathin sections, we observed an altered cell shape with reduced cell height and an increased cell width in MDCK_∆TTL_ cells (Figure 6A,B). The lack of TTL further impaired brush border membrane architecture as MDCK_∆TTL_ cells were characterized by a reduced number and altered shape of microvilli (Figure 6A,B).

This observation was confirmed by electron microscopical investigation of an ultrathin frozen cryosection (Figure 6C,D). Using the cryoimmunogold method, we observed a significant labelling of detyr-tubulin in MDCK_∆TTL_ cells, whereas this specific tubulin was nearly absent in wild-type MDCK cells (Figure 6C,D). We further evaluated the morphology of microvilli in MDCK_∆TTL_ cells and observed a significantly reduced length (0.59 ± 0.04 vs. 0.26 ± 0.05 µm, *** *p* < 0.001; Figure 6E) and number of microvilli (2 ± 0.2 vs. 1 ± 0.1 MV/µm, ** *p* < 0.01; Figure 6F) in the brush border membrane. Those results further strengthened our hypothesis that an altered tyrosination status of α-tubulin also affected the architecture of microvilli.

## 3. Discussion

Morphological changes of enterocytes, including a reduced cell height as well as an altered shape and number of microvilli, are recognized features of CD and are probably associated with intestinal malabsorption. The first descriptions of the altered morphology of enterocytes in CD date back to the 1960s, and recent studies were able to confirm those results [12,27]. One major characteristic of CD enterocytes is the reduced cell height, which is considered to be an early sign of CD inflammation in response to gliadin stimulation. Frisoni et al. showed that the reduction of CD enterocyte height already appears after 24 h gliadin treatment, which further strengthens the potential of this marker as a diagnostic tool [28]. However, the question arises as to whether those morphological alterations are rather a consequence of the inflammation or an inherent feature of CD. Mohamed et al. showed that enterocyte morphology was altered in untreated CD patients, but after introduction of the gluten-free diet, several parameters improved [27]. Sowinska et al. and Mohamed at al. further investigated enterocyte and microvillus morphology in patients with so-called potential CD, who already showed signs of CD autoimmunity (e.g., positive CD serology) but lacked the characteristic histopathological signs of overt disease. Both groups demonstrated that changes of enterocyte morphology already existed in patients with potential CD, which further indicates that those parameters might be helpful for an early diagnosis of CD [10,27].

Our own results showed a flattened morphology of enterocytes in CD epithelium that confirmed the abovementioned data. Using electron microscopy, we further observed changes of the architecture of the brush border membrane. However, to date, the molecular mechanisms leading to the observed morphological alterations have been hardly addressed. Microtubules represent one major element of the cytoskeleton, with multiple functions essential for cell shape, migration, and intracellular (biosynthetic as well as endocytotic) transport. Hence, we investigated the posttranslational modification of tubulin as a possible cause of the morphological alteration in CD enterocytes.

First, we demonstrated an increased amount of detyr-tubulin accompanied by a reduced cell height in CD enterocytes using immunofluorescence microscopy. As soluble tubulin heterodimers are rapidly tyrosinated by TTL [14], the detected detyr-tubulin must be present in polymerized microtubules. Those microtubules are considered as stable and less dynamic. Next, we applied cryoimmunogold electron microscopy to achieve a more detailed insight into the subcellular distribution of detyr-tubulin in the enterocytes. Again, we observed an increased concentration of detyr-tubulin in CD enterocytes, and its detection was most prominent in the apical region of enterocytes just below the terminal web. This observation was in line with our previous results, where we reported a decrease in the amount of detyr- and tyr-tubulin along the apico-basal axis in MDCK cells [29]. To further investigate the differential PTM of tubulin in CD, we quantified the protein levels of several tubulins and TTL using immunoblot. We observed an increased amount of the stable acetyl- and detyr-tubulin in newly diagnosed CD patients. The high levels of detyr-tubulin corresponded to a reduced expression of TTL in active CD, which was in line with our previous data obtained in MDCK cells and intestinal organoids [24,26]. However, the levels of TTL and detyr- and acetyl-tubulin were normalized in CD patients on a gluten-free diet, which indicated that the differential PTM of tubulins observed in active CD was rather mediated by early inflammatory mechanisms, which corresponded with observations from other groups [10,27]. Hence, the altered tyrosination status of microtubules could represent an early sign of the developing villous atrophy in CD.

Since other studies have described morphological changes of enterocytes in potential CD [10,27], one might assume that the pattern of posttranslational modification of tubulin is already disturbed in this stage.

To further elucidate the pathogenic mechanisms of altered PTM of tubulin in CD, we treated Caco-2 cells with recognized inflammatory mediators of the disease. First, we applied digested gliadin, which contains immunogenic and toxic gliadin peptides of α-gliadin. On the protein level, we observed a dose-dependent increase of the amount of detyr- and acetyl-tubulin. This effect was specific for gliadin, as we did not see any significant change in the level of detyr- and acetyl-tubulin after treatment with ovalbumin, which we used as a control food antigen. In line with this, the expression of VASH2, which is known to perform detyrosination of tubulin, was significantly increased in FF-treated but not in OVA-treated Caco-2 cells. We confirmed those observations by immunofluorescence microscopy. Again, we observed higher levels of detyr-tubulin and a reduced expression of TTL after treatment with digested gliadin. Strikingly, the altered PTM of tubulins was accompanied by a flattened morphology of Caco-2 cells. This altered cell shape after gliadin treatment matched the morphological features of enterocytes observed in CD biopsies [28]. Regarding the effect of gliadin, our results endorsed the recent observation of Porpora et al., who showed that the gliadin peptide P31–43 triggers an inflammatory response in CD organoids [30]. Next, we investigated the effect of IFN-y, the key cytokine of the Th1-mediated inflammation of CD. After 48 h treatment with IFN-y, we found an increased amount of detyr- and acetyl-tubulin. Furthermore, the expression of VASH2 was significantly increased.

Finally, we performed immunofluorescence microscopical imaging and observed again a reduced TTL expression, corresponding higher levels of detyr-tubulin, and an altered cell shape.

This prompted us to investigate whether a knockout of TTL would cause similar results. Hence, we investigated the subcellular distribution of detyr-tubulin and the morphology of the brush border membrane in wild-type and MDCK_∆TTL_ cells. The efficacy of TTL knockout and the corresponding increase in detyr-tubulin levels were shown by immunoblot and immunofluorescence in our previous publications [24,26]. On the electron microscopical level, detyr-tubulin was almost absent in wild-type MDCK cells, whereas we observed a strong labelling in MDCK_∆TTL_ cells, predominantly in the apical region. MDCK_∆TTL_ cells further displayed a flattened morphology [24], as seen in CD enterocytes and Caco-2 cells after stimulation with digested gliadin and IFN-y. Strikingly, the number and architecture of microvilli were dramatically altered in cells lacking the enzyme TTL. Thus, reduced expression of TTL, which results in increased intracellular quantities of detyr-tubulin, seemed to affect not only the cell shape but also the architecture of the brush border membrane.

It is known that microtubules do not only influence cell shape but also play a diverse role in transport processes as depolymerization by colchicine and vinblastine significantly impairs the polarized transport of brush border membrane enzymes (sucrase isomaltase, alkaline phosphatase, and aminopeptidase N) and alters the localization of key elements of apical microvilli (actin, fimbrin, and villin) [31]. Evidence for a link between microtubule-dependent transport and the biogenesis of microvilli also comes from studies on Drosophila melanogaster epithelia [32,33]. Khanal et al. found that endosomes are transported along microtubules to deliver Cadherin 99C, which is sufficient to expand microvilli length, to the apical domain of epithelial cells [31]. Recent data further suggest a direct interaction between α-tubulin and actin [34,35]. Reinke et al. demonstrated significant changes in the intracellular distribution of actin and tubulin after stimulation of Caco-2 cells by digested gliadin. Increasing doses of digested gliadin reduced the amount of actin in the brush border membrane, whereas the level of tubulin was increased. Those observations were associated with impaired apical trafficking of brush border membrane enzymes [36]. In addition, altered transport kinetics depending on the tyrosination status of microtubules might play a role as well. In vitro data from Sirajuddin et al. indicate a differential binding and velocity of kinesin-1 and -2 on detyr-tubulin, and our own previous data have shown that modulation of TTL expression by MDCK cells alters apical transport routes of the secretory glycoprotein 80 and the membrane-associated protein 75 [29]. In the context of our results, we assume that microtubule-associated transport of brush border membrane proteins is disturbed by an altered tyrosination status of tubulin in CD and results in a reduced density, disturbed morphology, and potentially impaired function of the brush border membrane (Figure 7A and B). The altered expression of TTL and VASH2 in CD might thus contribute to malabsorption by altering the architecture of the intestinal brush border membrane.

In summary, our data are the first to elucidate the underlying functional mechanisms for the morphological changes of enterocytes in CD. CD-associated inflammation induced by digested gliadin and IFN-y leads to a reduced expression of TTL and a higher expression of VASH2 resulting in increased amounts of stable detyr-tubulin. This altered tyrosination of microtubules does not only influence cell shape but also disturbs brush border membrane architecture. The observed effects seem to be CD-specific since ovalbumin did not induce any changes in the PTM of microtubules. Further, the tyrosination status of α-tubulin was restored after initiation of a gluten-free diet. The disturbed posttranslational modification of microtubules might contribute to CD-associated villous atrophy and potentially even malabsorption. Future research should aim to investigate the potential of our findings, for example, in the diagnosis of potential CD.

## 4. Materials and Methods

### 4.1. Patients’ Characteristics

The study was approved by the local ethics committee (reference number 246/16), and written informed consent was obtained from every family. The study was performed in accordance with the ethical principles from the Declaration of Helsinki and with the local ethic guidelines. Duodenal biopsies were obtained during clinically indicated upper gastrointestinal endoscopy. Patients with active CD and non-celiac disease (control) patients were on a gluten-containing diet (Table 1). The effect of the gluten-free diet was analyzed by immunoblots of biopsies from CD patients on a gluten-free diet for at least 6 months.

Tissue samples from nine CD (mean age: 11 (3–16) years, six females) patients were obtained. All patients showed clinical signs of CD and a typical histopathology (MARSH III a–c). Except for one patient with selective IgA-deficiency, all patients had a positive serology (anti-transglutaminase 2-IgA antibodies: 176 (20–200) IU/mL). The diagnosis of CD was made according to current guidelines of the European Society for Paediatric Gastroenterology, Hepatology and Nutrition (ESPGHAN) [37]. In addition, biopsies from 10 control patients (mean age: 13 (8–16) years, seven females) on a gluten-containing diet were obtained. The control group consisted of one patient with eosinophilic esophagitis, one patient with familial adenomatous polyposis, two patients with functional abdominal pain, two patients with Crohn’s disease, two patients with type c gastritis, and two patients with reflux esophagitis. None of the control patients had any endoscopic or histopathologic sign of duodenal inflammation. CD serology was negative in all control patients. In addition, duodenal biopsies from four CD patients (8 (3–16) years, all female) on a gluten-free diet for at least 6 months were obtained for immunoblot analysis. Patients underwent endoscopy because of persistent symptoms and positive serology (anti-transglutaminase 2-IgA antibodies: 98 (20–200) IU/mL, MARSH 0-IIIa).

### 4.2. Cell Culture

Human intestinal epithelial cells (Caco-2) were cultured at 37 °C with 5% CO2 and 95% humidity in DMEM (Dulbecco’s Modified Eagle’s Medium, Gibco) supplemented with 1% penicillin-streptomycin, 1% essential amino acids, 1% sodium pyruvate, and 10% heat-inactivated fetal bovine serum. The culture medium was changed every two to three days, and cells were passaged at 80% confluence using trypsin-EDTA. Cells were seeded at a density of 4.5 × 10^4^ cells per cm². For this study, Caco-2 cells were used at passages 10–50 and 7–14 days after confluency. Investigation of the detyrosination of Caco-2 cells was done after stimulation with IFN-y (SRP3058, Sigma-Aldrich, Darmstadt, Germany), ovalbumin (Sigma-Aldrich, Darmstadt, Germany), or digested gliadin (Frazer’s Fraction III) [38] for 48 h at the indicated doses.

MDCK type II and MDCK_∆TTL_ cells were cultured at 37 °C with 5% CO2 in Minimum essential medium (MEM; Gibco; Langenselbold, Germany) supplemented with 2 mM glutamine, 5% fetal calf serum, 100 U/mL penicillin, and 100 mg/mL streptomycin. For the generation of the MDCK_∆TTL_-cell line, TTL expression was eliminated by CRISPR/Cas9 using the plasmid pSpCas9n(BB)-2A-Puro (PX462) V2.0, which was a gift from Feng Thang (Addgene plasmid #62987). Oligo pairs encoding the 20 nt guide sequences against canine TTL (5′-CAC CGA ATA TCT ACC TCT ATA AAG A-3′; 5′-AAA CTC TTT ATA GAG GTA GAT ATT C-3′) were annealed and ligated into the BbsI digested plasmid to generate pCRISPR-Cas9 ΔTTL. Following transfection of pCRISPR-Cas9 ΔTTL with Lipofectamine2000, cells were selected for 48 h with 2 µg/mL puromycin (Sigma-Aldrich, Darmstadt, Germany). Lysates of MDCK cell clones were analyzed for the presence of TTL by immunoblot using antibodies directed against TTL. Clones devoid of TTL expression were selected. The efficient knockout of TTL and the corresponding increase in detyr-tubulin levels have been described in our previous publications [24,26].

### 4.3. Immunoblot

For immunoblot of duodenal biopsies, samples were washed in PBS (phosphate buffered saline) and lysed in 200 µL RIPA lysis buffer supplemented with phenylmethylsulfonyl fluoride, protease inhibitor, and sodium orthovanadate (1% each) for 20 min on ice and centrifuged (20 min at 14,500 rpm at 4°C) afterward. For immunoblot of Caco-2 cells, cells were washed two times with ice-cold PBS, scraped, and centrifuged (60 s at 14,500 rpm). Supernatant was collected, and cell debris was discarded. Protein quantitation was performed using Pierce BCA protein assay kit (Thermo Fisher Scientific, Langenselbold, Germany). After denaturation with Laemmli buffer containing 2-mercaptoethanol for 5 min at 95°C, 20 µg of the protein extract was loaded per lane on a 10% SDS-gel. Electrophoresis was done for 1 h at 40 mA. Protein transfer onto a nitrocellulose membrane (0.2 µm pore size) was done using a Transblot Turbo system (Bio-Rad Laboratories, Feldkirchen, Germany) for 30 min (25 V, 1 A). The membrane was blocked for 1 h with 5% dry milk in PBS + 2.5% Tween. Primary antibodies (Table 2) were applied overnight at 4 °C at indicated dilutions in the blocking buffer. After three washes with PBS + 2.5% Tween, the membrane was incubated with horseradish peroxidase-conjugated secondary antibodies (1–1000, Santa Cruz Biotechnology, Heidelberg, Germany) for 1 h. The membrane was washed again three times with PBS + 2.5% Tween and developed using the SuperSignal West Femto kit (Thermo Fisher Scientific) for 5 min. Image processing and quantitation were performed using a ChemiDoc XRS+ imager and Image Lab software (Bio-Rad Laboratories).

### 4.4. Immunofluorescence Microscopy

For immunofluorescence analysis, cells were grown on cover slips and fixed with 4% paraformaldehyde for 20 min. After fixation, the cells were permeabilized with 0.1% Triton-X-100 for 20 min and blocked in 5% BSA/PBS++ (PBS supplemented with 1 mM MgCl2 and 1 mM CaCl2) for 1 h. Immunostaining was performed with the corresponding primary antibodies (Table 1) in blocking reagent for 2 h. Specific secondary antibodies labeled with indicated Alexa Fluor dyes were applied in PBS++ for 1 h. Hoechst 33342 was used for nuclear staining. Following incubation, the cells were washed with PBS++ and mounted with ProLong Diamond (Thermo Fisher Scientific). Intestinal tissues were received from patients in the Department of Pediatrics, Justus-Liebig-University Giessen, for diagnostic purposes as described above. Four micrometer thick slices of formalin fixed and paraffin embedded human small intestinal samples were steamed in Tris/EDTA for 20 min and blocked in 5% goat serum/PBS. Primary and secondary antibodies were incubated in antibody diluent (Dako, Hamburg, Deutschland). Confocal images were acquired on a Leica STELLARIS equipped with a ×93 glycerol and a ×40 or ×63 oil plan-apochromat objective (Leica Microsystems, Wetzlar, Germany). Fluorescence images were recorded by excitation with 405 nm (Hoechst 33342), 488 nm (Alexa488, Thermo Fisher Scientific), 555 nm (Alexa 555), and 647 nm (Alexa 647) using the 404 nm laser or the white light excitation laser. For specific emission detection, the following wavelength ranges were selected in sequential scan mode: 460 to 490 nm (Hoechst 33342); 510 to 550 nm (Alexa 488); 570 to 600 (Alexa 555); and 660 to 680 (Alexa 647). The laser power set up, gain parameter, and pinhole settings were identical for each experiment (control cells and treated cells). All images were processed identically with Leica LAS X and exported for further analysis by ImageJ. The immunofluorescence intensities were evaluated per fluorescence channel (detyr-tubulin; TTL; tyr-tubulin). For this purpose, defined areas were analyzed in the perinuclear and peripheral regions of the cell and applied for each single channel. The analysis was based on the mean value for the gray level intensity of the respective channel. For the analysis, at least nine images (three to seven cells per image) from three independent experiments were analyzed by ImageJ.

### 4.5. Electron Microscopy

Duodenal biopsies were sectioned and labelled using the postembedding technique according to Tokuyasu et al. [39] and Griffiths [40]. In brief, biopsies were washed three times in PBS and fixed for 1 h in 5% paraformaldehyde-PIPES buffer at room temperature afterward. Then, samples were placed in polyvinylpyrrolidone-sucrose at 4 °C overnight and were cryopreserved in liquid nitrogen the next day. For cryoimmunogold electron microscopy of cell lines, cells were fixed with 4% paraformaldehyde at room temperature for 30 min. After that, the cells were washed with PBS and embedded in gelatine at 37 °C, and after cooling down to 4 °C, samples were placed in polyvinylpyrrolidone-sucrose at 4 °C overnight and cryopreserved in liquid nitrogen the next day. For preparation of epon sections for electron microscopy, cells were fixed in 1% glutaraldehyde in 0.1 mol/L sodium Cacodylate buffer (pH 7.2, 1 h, room temperature), postfixed with 2% osmium tetroxide (OsO4), and contrasted with 0.5% uranyl acetate overnight. Samples were dehydrated by a series of graded ethanol washes and embedded in epon. To examine cellular morphology, ultrathin sections (60–80 nm) of epon-embedded samples were produced, placed onto mesh copper grids, and stained with lead salts. For cryoimmunogold electron microscopy, ultrathin frozen tissue sections (55 nm) were produced and incubated with blocking solution (0.1% cold water fish skin gelatine, 5% fetal calf serum, and 5% goat serum) for 15 min at room temperature. After incubation with the primary antibody (Table 1) in blocking solution for 45 min at room temperature, samples were washed five times in PBS, followed by incubation with the secondary gold-conjugated goat sera against rabbit immunoglobulin G (1:40, 12 nM, Jackson ImmunoResearch) in blocking buffer for 45 min. After washing in PBS, contrasting was carried out using uranyl acetate and 2% methylcellulose.

Imaging was performed using a transmission electron microscope (Philips EM 400 T, Kassel, Germany). For quantitative studies, five photographs of each sample were taken by a blinded observer at a magnification of 17,700×, and photographs were manually developed and scanned at 600 dpi. Image processing was done with Image J [37,41]. Labelling of cytosolic detyrosinated tubulin was quantified and background subtracted against labelling on the nucleus.

### 4.6. Statistics

Statistical analysis was performed using SPSS 21 (IBM) and GraphPad Prism 9 (GraphPad Prism Software Inc., San Diego, CA, USA). Results were tested for normal distribution. Students unpaired two-tailed *t* test (with Welch correction where appropriate) or U test were used where appropriate. *p* values less than 0.05 were considered significant.

## Figures and Tables

**Figure 1 ijms-24-02635-f001:**
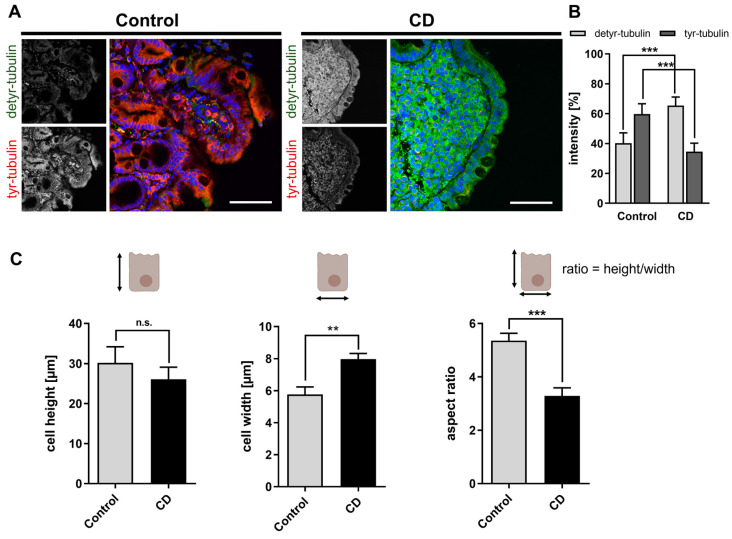
Altered tyrosination of α-tubulin and disturbed morphology of enterocytes in CD. (**A**) Tissue sections of duodenal biopsies from control and CD patients were stained with antibodies against detyr- (Alexa Fluor 488, green) and tyr-tubulin (Alexa Fluor 647, red). Images were taken with a 40× oil objective. Nuclear counterstaining with Hoechst 33342 is depicted in blue. (**B**) Detyr- and tyr-tubulin intensities were quantified using the ImageJ measurement tool. Intensity ratios are shown for both groups of patients. The level of tyr-tubulin was significantly decreased in CD biopsies in favor of increased amounts of detyr-tubulin. (**C**) Quantitation of cell height, width, and aspect ratio (height/width). CD enterocytes display a flattened morphology with reduced height but increased width. Data are shown as mean ± SD, *n* = 3. Statistical significance was tested with Student’s *t*-test (n.s. is not significant; ** *p* < 0.01; *** *p* < 0.001). Scale bars: 50 µm.

**Figure 2 ijms-24-02635-f002:**
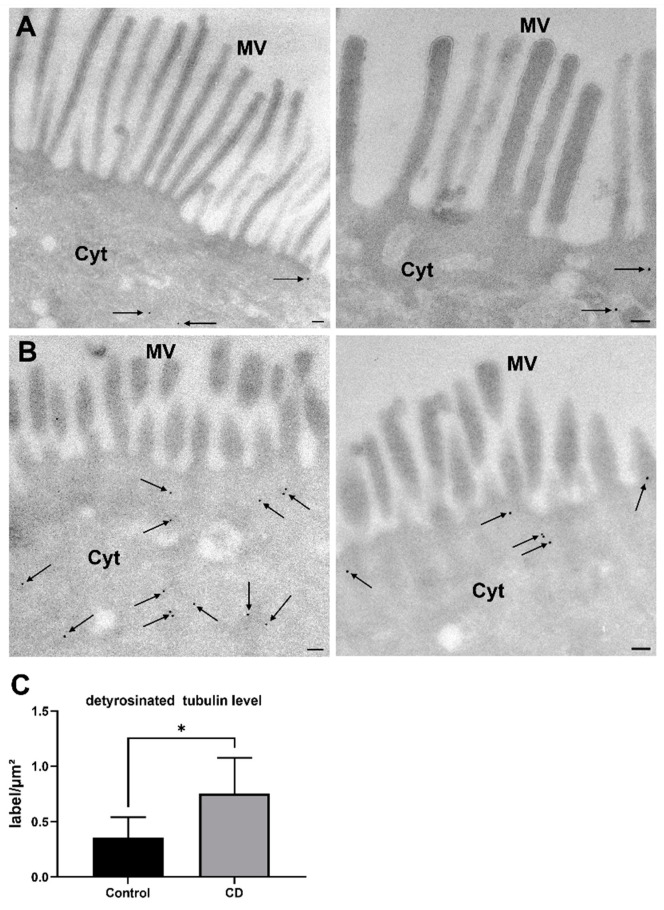
Increased prevalence of detyr-tubulin and altered morphology of microvilli in CD biopsies. Ultrathin frozen cryosections (55 nm) of duodenal biopsies from control (**A**) and CD (**B**) patients were stained with a primary polyclonal antibody directed against detyr-tubulin, and visualization was done using a species-specific, gold-conjugated secondary antibody (goat-anti-rabbit, 12 nm gold particles, arrows). After contrasting, blinded acquisition of standardized images was done. (**C**) Cytosolic detyr-tubulin levels were quantified using the ImageJ measurement tool. Cytosolic detyr-tubulin quantities were increased in CD enterocytes. Images were taken with a transmission electron microscope at 17,700-fold magnification. Data are shown as mean ± SD, *n* = 5. Statistical significance was tested with Student’s *t*-test (* *p* < 0.05). Scale = 0.1 µm. Cyt, cytosol; MV, microvilli.

**Figure 3 ijms-24-02635-f003:**
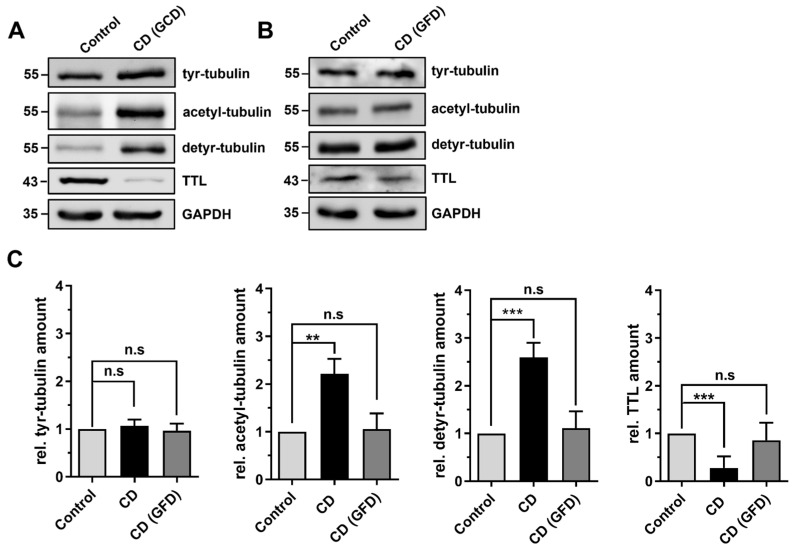
Protein levels of posttranslationally modified tubulins in duodenal biopsies of CD patients with active disease and in remission. (**A**,**B**) Equal amounts (20 µg) of protein lysates obtained from duodenal biopsies were analyzed by immunoblot with antibodies directed against detyr-, tyr-, and acetyl-tubulin, TTL, and GAPDH. Samples of patients with newly diagnosed CD on a gluten-containing diet (GCD) (**A**) as well as patients with CD on a gluten-free diet (GFD) (**B**) were compared with control patients. GAPDH served as a loading control. (**C**) Relative protein expression was normalized to GAPDH levels. Quantities from the control group were set as 1. There was a significantly increased amount of acetylated and detyr-tubulin and reduced expression of TTL in active CD. After introduction of a gluten-free diet, the expression patterns of modified tubulins were in the range of control biopsies. Data are shown as mean ± SD, *n* = 3. Statistical significance was tested using one-way ANOVA with Dunnet’s comparison (n.s. is not significant; ** *p* < 0.01; *** *p* < 0.001). GFD, gluten-free diet; TTL, tubulin tyrosine ligase.

**Figure 4 ijms-24-02635-f004:**
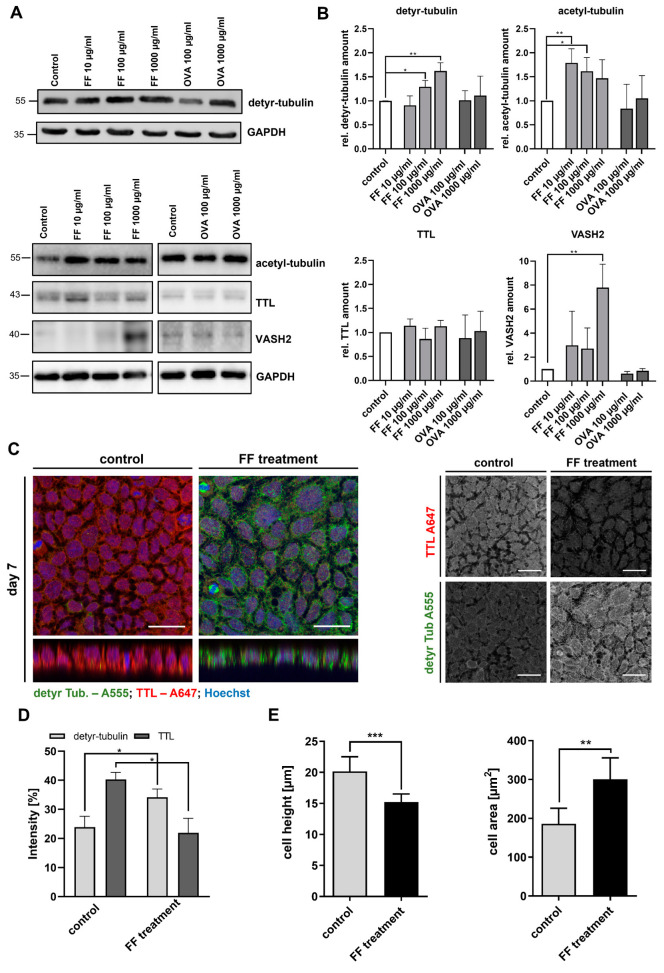
Stimulation of Caco-2 cells with digested gliadin influences tyrosination status of cellular tubulin and alters cell shape. (**A**) Cellular quantities of detyr- and acetyl-tubulin, TTL, and VASH2 in Caco-2 cells were analyzed by immunoblot after treatment with increasing amounts of FF (10–1000 µg/mL) or ovalbumin (100–1000 µg/mL). GAPDH served as loading control. (**B**) Relative quantities were normalized to GAPDH levels in whole cell lysates. Quantities from the control group were set as 1. There was a dose-dependent increase in detyr-tubulin and acetyl-tubulin levels and increased VASH2 expression after treatment with FF. Incubation with ovalbumin as control antigen did not alter the PTM of tubulin. (**C**) Presence of detyr-tubulin (Alexa Fluor 488, green) and TTL (Alexa Fluor 647, red) in confluent monolayers of differentiated Caco-2 cells were analyzed after stimulation with FF (1000 µg/mL) by confocal microscopy. FF treatment reduced TTL expression but increased levels of detyr-tubulin. Images show the reduced cell height (z-stack) and the increased cell area of Caco-2 cells after FF treatment. Images were recorded with a 40× oil objective. Nuclear counterstaining with Hoechst 33342 is depicted in blue. Scale bars: 30 µm. (**D**,**E**) Quantitation of detyr-tubulin and TTL staining in treated Caco-2 cells. Intensity as well as cell area and height were measured by the ImageJ analysis tool. The Caco-2 cell height was reduced, whereas cell area increased after treatment with FF. Data are shown as mean ± SD, *n* = 3. Statistical significance was tested with Student’s *t*-test (* *p* < 0.05; ** *p* < 0.01; *** *p* < 0.001). FF, Frazer’s Fraction III; OVA, ovalbumin.

**Figure 5 ijms-24-02635-f005:**
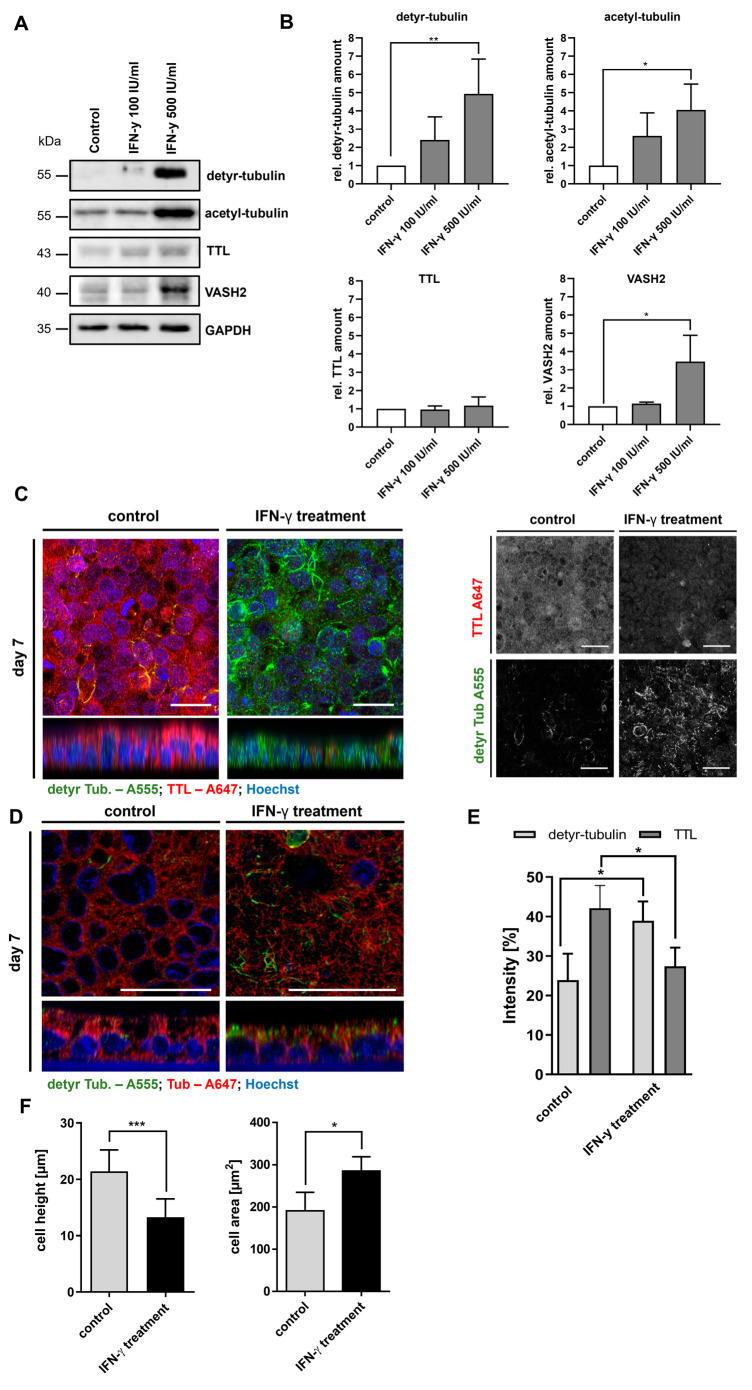
IFN-y treatment increases detyr-tubulin and acetyl-tubulin levels and alters cell shape of Caco-2 cells. (**A**) Cellular quantities of detyr- and acetyl-tubulin and expression of TTL and VASH2 were analyzed by immunoblot after stimulation of Caco-2 cells with different doses of IFN-y for 48 h. GAPDH served as loading control. (**B**) Relative quantities were normalized to GAPDH levels in whole cell lysates. Quantities from control group were set as 1. There was a dose-dependent increase in detyr- acetyl-tubulin levels and VASH2 expression after treatment with IFN-y. (**C**,**D**) Presence of detyr-tubulin (Alexa Fluor 488, green) and TTL (Alexa Fluor 647, red) or tubulin (Alexa Fluor 647, red) in confluent monolayers of differentiated Caco-2 cells were analyzed after stimulation with IFN-y by confocal microscopy. IFN-y treatment reduced TTL expression but increased levels of detyr-tubulin. Detyr-tubulin enriched microtubules were predominantly detected in subapical areas above the cell nuclei. Images show the reduced cell height (z-stack) and the increased cell area of Caco-2 cells after IFN-y stimulation. Images were recorded with a 40× oil objective. Nuclear counterstaining with Hoechst 33342 is depicted in blue. Scale bars: 30 µm. (**E**,**F**) Quantitation of detyr-tubulin and TTL staining in treated Caco-2 cells. Intensity as well as cell area and height were measured by the ImageJ analysis tool. Caco-2 cell height was reduced, whereas cell area increased after treatment with FF. Data are shown as mean ± SD, *n* = 3. Statistical significance was tested with Student’s *t*-test (* *p* < 0.05; ** *p* < 0.01; *** *p* < 0.001).

**Figure 6 ijms-24-02635-f006:**
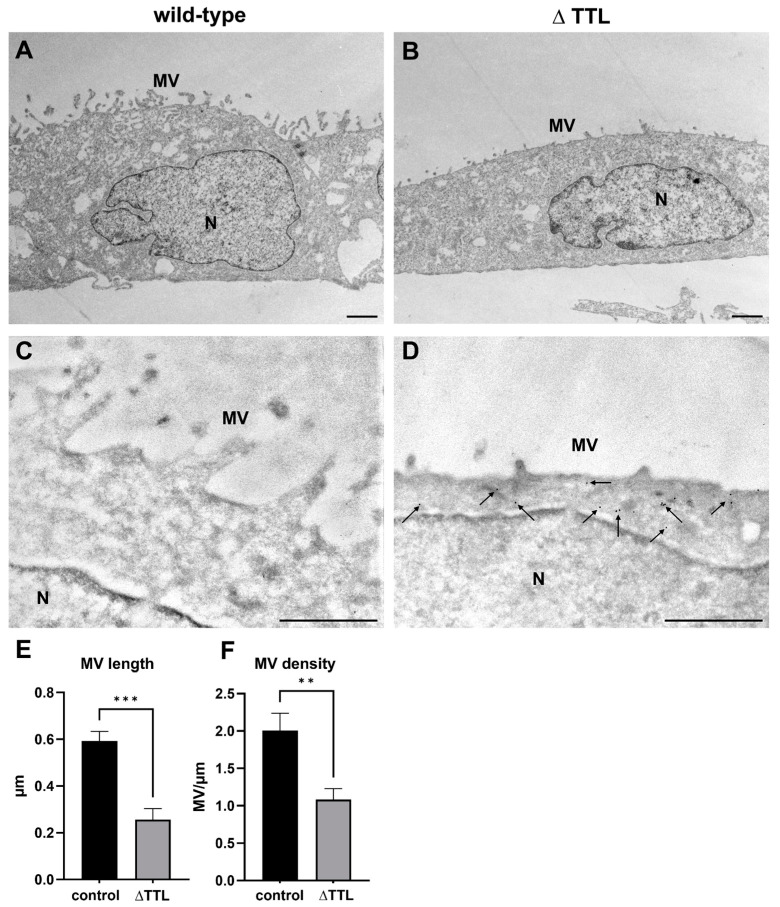
Altered morphology of microvilli and increased amount of detyr-tubulin in MDCK_∆TTL_ cells. Ultrathin sections of epon-embedded wild-type (**A**) and MDCK_∆TTL_ cells (∆ TTL) (**B**). Detyr-tubulin detection in ultrathin cryosections of wild-type (**C**) and MDCK_∆TTL_ cells (**D**). Knockout of TTL results in an altered cell and microvillus shape associated with increased presence of detyr-tubulin (D, 12 nm gold particles, arrows). (**E**) Reduced length of microvilli in MDCK_∆TTL_ cells. (**F**) Reduced number of microvilli per µm of apical membrane in MDCK_∆TTL_ cells. Images were taken with a transmission electron microscope at 4400-fold (**A**,**B**) and 17,700-fold (**C**,**D**) magnification. Quantitation of microvillus height and density was performed with ImageJ. Scale = 1 µm. Data are shown as mean ± SD, *n* = 3. Statistical significance was tested with Student’s *t*-test (** *p* < 0.01; *** *p* < 0.001). MV, microvilli.

**Figure 7 ijms-24-02635-f007:**
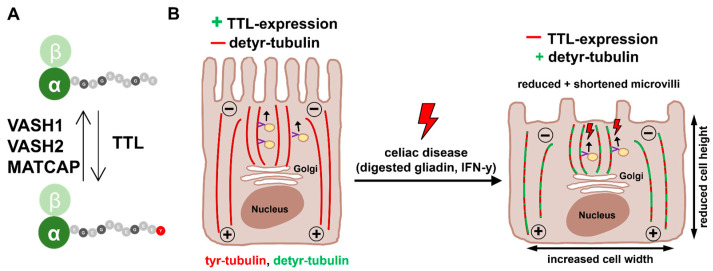
Pathogenesis of PTM of α-tubulin and morphological changes in CD enterocytes. (**A**) Tubulin tyrosine ligase (TTL) specifically catalyzes tyrosination of detyr-tubulin heterodimers. Detyrosination of tubulin is mediated by VASH1, VASH2, and MATCAP. (**B**) In regular enterocytes, TTL is strongly expressed, causing low levels of detyr-tubulin. Inflammatory triggers in CD such as digested gliadin or IFN-y reduce TTL expression resulting in significantly increased amounts of detyr-tubulin. Alterations of the tyrosination status of microtubules are accompanied by an altered cell shape (e.g., reduced cell height and increased cell width). Reduction of TTL expression also leads to a reduced number of shortened microvilli. Impaired brush border membrane architecture may be caused by an altered microtubular transport of proteins from the Golgi apparatus to the brush border membrane and/or modified interaction between microtubules and the actin cytoskeleton.

**Table 1 ijms-24-02635-t001:** Patients’ characteristics.

Patient	Patient Group	Age	Sex	Anti-TG2-IgA (IU/mL)	MARSH	Remarks
1	CD	10	f	20	3a-b	
2	CD	16	f	200	3b	
3	CD	14	m	11	3b-c	IgA deficiency
4	CD	13	f	200	3a-b	
5	CD	11	m	200	3b-c	
6	CD	3	f	200	3b	
7	CD	6	f	200	3a	
8	CD	10	f	186	3b	
9	CD	13	m	200	3a	
10	CD GFD	4	f	62	2	
11	CD GFD	16	f	20	0	
12	CD GFD	9	f	34	0	
13	CD GFD	3	f	200	3a	
14	Co	15	f	<20	0	Functional abdominal pain
15	Co	15	f	<20	0	Familial adenomatous polyposis
16	Co	8	m	<20	0	Reflux esophagitis A
17	Co	14	f	<20	0	Reflux esophagitis A
18	Co	13	m	<20	0	Eosinophilic esophagitis
19	Co	11	f	<20	0	Crohn’s colitis
20	Co	14	f	<20	0	Gastritis type C
21	Co	16	f	<20	0	Gastritis type C
22	Co	11	m	<20	0	Crohn’s colitis
23	Co	10	f	<20	0	Functional abdominal pain

**Table 2 ijms-24-02635-t002:** Primary antibodies used in this study.

Target	Antibody	Host/Clonality	Dilution	Company
Acetylated α-tubulin	T6739	Mouse/monoclonal Clone 6-11B-1	IB: 1–500	Sigma-Aldrich, St. Louis, MO, USA
Detyrosinated α-tubulin	#AB3201	Rabbit/polyclonal	IB: 1–5000 IF: 1–200 EM: 1–10	Merck Millipore, Darmstadt, Germany
GAPDH	5G4	Mouse/monoclonal	IB: 1–500	HyTest Ltd., Turku, Finland
Tubulin tyrosine ligase	13618-1-AP	Rabbit/polyclonal	IB: 1–500 IF: 1–200	Proteintech, Rosemont, IL, USA
Tyrosinated α-tubulin	Sc-53029	Rat/monoclonal Clone YL1/2	IB: 1–5000 IF: 1–200	Santa Cruz Biotechnology, Dallas, TX, USA
VASH2	ab224723	Rabbit/polyclonal	IB: 1–2000	Abcam, Cambridge, UK

IB, immunoblot; IF, immunofluorescence microscopy; EM, electron microscopy.

## Data Availability

The data presented in this study are available on request from the corresponding author.

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
