# Peer review of "Altered Posttranslational Modification of Microtubules Contributes to Disturbed Enterocyte Morphology in Celiac Disease"

_ijms, 2023, doi:10.3390/ijms24032635_

Round 1

Reviewer 1 Report

The Manuscript "Altered posttranslational modification of microtubules contributes to 

disturbed enterocyte morphology in celiac disease" by Stircker Sebastian et al. focus on structural alterations, namely microtubules structure and composition, in CD enterocytes. A secondary epithelial cell line such as MDCH lacking TTL has similar alterations of the c ell shape and microvilli founding CD biopsies.Moreover Gliadin peptides induced in another secondary epithelial cell line such as caco2 cells similar modicications of the TTL. The Gliadin peptides effect was transient.  The results presented in this manuscript indicate that gliadin peptides can induce alterations similar to those found in CD cells without treatment.  

This is a novel finding that confirm similarly observations published on IJMS, done on CD intestinal epithelial cells using other read outs Porpora M et al Inflammation Is Present, Persistent and More Sensitive to Proinflammatory Triggers in Celiac Disease Enterocytes IJMS 2022 Feb 10;23(4):1973. doi: 10.3390/ijms23041973.]

This should be addressed in the discussion section. 

Other minor points to be addressed:

  1. the figure in the introduction is not necessary, can be moved to the discussion 

Moreover, enterocytes of 72 intestinal organoids distributed along the crypt villus axis in the small intestine also ex- 73 hibit TTL-dependent changes in cell shape 

  1. Line 72-73. “Enterocytes of intestinal organoids distributed along the crypt-villus axis” is this correct? Could be “biopsies” instead of “organoids”? 
  2. Line 96. “Abundance”? The authors mean “amount”?
  3. In figure 4 GFD is missing from the B section of the figure
  4. In the material and methods section a table of the patients and their main parameters (TTG, Marsh stage, age, and sex) is missing

Reviewer 2 Report

Duodenal mucosa in inflammatory celiac disease (CD) patients present an increase of CD3+ intraepithelial lymphocytes and villous atrophy. In addition, change of enterocytes morphology, especially the brush border membrane has also, for long, been observed in CD patients.

This paper is addressing the potential importance of a microtubule (MT) post translational modification, ie: detyrosination, on the poorly understood, morphological changes that occur in enterocytes cells during CD and makes a first link between the two. The approach is based on the analyses of healthy and CD patient biopsies and on the manipulation of Caco-2 and MDCK cells, in order to mimic cell morphological changes as observed in enterocytes of CD patients.

The paper is interesting and well written with a good introduction to CD and discussion. The experimental approach undertaken is quite straightforward and convincing (but Figure5). Nonetheless, a couple of points should be addressed to strengthen the paper prior its publication.

In summary, at the beginning of the paper, the authors not only analyze the pattern of MT detyrosination but also of MT acetylation in CD biopsies. Although changes in MT acetylation behaviour are also observed, they later on focus on MT detyrosination. It would have been interesting to see whether MT Acetyl tubulin level is also affected in Caco-2 treated cells. The authors have had access to a definite number of biopsies from healthy and CD patients, but western blots results are given from only two patients. I am not very convinced by the gliadin results.

Finally, the authors observation of the change of PTM pattern in CD biopsies or Caco-2 stimulated cells does not imply that the MT detyrosination is responsible for the cell shape change. MT detyrosination and morphological changes could be independent consequences of CD or cells stimulation. To prove that point, link the two events and show that MT detyrosination does indeed induce cell shape change, the authors compare the brush border membrane architecture of naïve and TTL minus MDCK cells. They provide interesting results and the ultrastructure shown is very convincing. Nevertheless, controls are missing on the presented Data of MDCK cells. In addition, the results shown could also be strengthen by the overexpression of VASH1/2

Detailed Points:

Detyr/Tyr MT ratio, Figure 2

Comparison of Detyr/Tyr tubulin ratio in tissue section from duodenal biopsies are very nice.  The n=3 in Figure 2 means 3 biopsies from CD or healthy patients were analyzed??

I am not an expert in CD and I am not sure whether patients at different stages of the disease can be identified. If it was the case, it would be interesting to compare these ratio at different stages of the disease. Would that be possible?

Detyr/Tyr MT ratio, Figure 4

The authors compare levels of MT PTMs from disease free individuals with CD patients (untreated) and CD patients on a gluten free diet biopsies. The changes observed in panel A are striking but we are looking at the results obtained from only two patients biopsies. Why not show, side by side, the full panel of tissue samples that were available to the authors (9 CDs and 10 controls according to the experimental procedure). The full set would be much more convincing.

How are the biopsies made/ is it possible that other tissue might be contaminating??

Same comment than for Figure 2: To ascertain these results, it would also be nice to compare protein levels in two different biopsies from the same patient realized before treatment and after gluten free treatment.

Results from gliadin treatment: Figure 5

The results showing western blots of gliadin stimulated Caco-2 cells are puzzling (Figure 5).

 Indeed, western blots shown and quantification made from them appear conflicting. I know that the human eye can only distinguish a couple of hundred of grey levels.

 Nonetheless, the authors claim that exposure of cells to increasing concentration of digested gliadin results in an increase of Detyr tubulin. This is really hardly visible (Figure 5 A). But what annoys me most is that the lane 5 of the WB (Detyr) showing exposure to ovalbumin is much fainter than in control while lane 6 is much brighter. While levels of GAPDH look similar to me, quantification in Panel B claims that the relative amount of Detyr tubulin is similar in ovalbumin treated cells (sample 5 and 6) than in control. Did I miss something?

- Why not use also the TTL antibody on western blot, that would strenghen the results.

- Western blots are also somehow contradictory with the IF shown in panels C/D. Indeed, the immunofluorescence shown is clear cut.

The Detyr MT level increases by an order of several folds between control and FF treated cells and not a mere 1.5x fold as claimed in the WB (Panels A and B).

Were the images shown in panel C acquired with an identical laser power set up for control and FF treated cells and were they further processed identically for figure mounting?

- Was the objective used really be x93 (same for figure 6)? A lot of cells are shown..    

- The way images were quantified should be better explained and not just refer to image J particle analysis tool. Were thresholded images in one channel, applied to the other channel for intensities measures?  How many cells were quantified?

- In Figure 6, it would be nice to show a high magnification of the MT network (Total and Detyr tubulin) in control and IFN-g treated Caco-2 cells, to better visualize the changes in MT organization.

- What about MT acetylation status in Caco-2 treated cells?.

Detyrosination of the MT network is responsible for cell shape changes.

To show the relationship between loss of tyrosinated MTs and change cell shape, the authors use naïve or TTL deleted MDCK cells (Figure 7). This is an interesting experiment and while, the electron microscopy shown is very nice, the experiment lacks some basic controls. It will be important to show western blot of TTL (if they cross react with dog specie) and Detyr MTs or at least, Tyr and Detyr MTs to confirm the efficacy of TTL deletion. In addition, immunofluorescence of the MDCK cells should be shown using TTL antibodies (if they cross react with dog specie) or at least detyrosinated and tyrosinated tubulin tubulin as was performed for the treated Caco-2 cells. Consequences of overexpressing VASH1/2 would add a plus

Minor points

- Introduction is clear. I would just add a couple of sentences concerning MT PTMs. Indeed, a lot more is known, today about their function, that what is stated in the introduction (lanes 62-63)..…MT Acetylation has well known functions in neural physiology and autophagy, while detyrosination of subspecies of MT is especially important for proper progression of mitosis… So please just cite a couple of reviews! !

- Figure 4 western blot panel

Panel A top legend should precise CD gluten Plus diet

Panel B top legend, CD gluten free diet is missing

- Sentence lane 202-204 is very long and should be cut in two sentences.

- “On the protein level, we observed a significant, dose-dependent in-205 crease in the cellular level of detyr-tubulin after 48 hours of treatment with 500 IU/ml IFN-206 y (4.9 ± 1.9, p < 0.01, Figure 5A and B). 

It is figure 6A and B

- It seems very unlikely, considering, the number of cells shown in Figures 5 panel C and 6 panel that the images were acquired with a x93 mag objective as indicated in the figure legends…Please check and edit.

- Z section shown in Figure 5C and 6C are nicely showing the cell shape change, but are not commented in the text. A sentence should be added as the results shown reinforce the point the authors want to make

Round 2

Reviewer 2 Report

The authors clarify a number of points in their answer. Once again, I find this is an interesting study.

I however insist that some time should be used to try to perform the very simple experiments, I asked for.

Regarding figure 5 now 4B: Please provide the original Data

“However, for the quantitation, all 3 experiments were evaluated, which is shown in revised figure 4b. If desired, we can provide the original data resulting from the densitometric analysis. “

Please perform the experiments

            Previous question was: Why not use also the TTL antibody on western blot, that would strenghen the results.  In the context of the revision of the manuscript we performed immunoblots regarding the TTL expression of Caco-2 cells after stimulation with FF, OVA and IFN-y. However, our primary investigations did not show clear effects regarding the TTL expression. For the detailed investigation of this issue, we would have to seed new Caco-2 cells and perform treatment of 7-14 day old cells with the mentioned substances again. Since the time period for the revision was limited, this was not feasible. However, if desired, we will perform additional experiments regarding this issue. “

I know that cells grown several days after confluency do not have a nice microtubule network.  The purpose of showing such micrograph is not to see a nice MT network, but eventually to see and analyze potential changes in its organization. I still think that high magnification of the MT network will be informative.

            Previous question was: In Figure 6, it would be nice to show a high magnification of the MT network (Total and Detyr tubulin) in control and IFN-g treated Caco-2 cells, to better visualize the changes in MT organization. 

- “We did not use high magnification images of Caco-2 cells since the visualization of the ultrastructural MT network was not satisfactory in the confluent polarized monolayers of Caco-2 cells. One reason for this is that unlike MDCK cells, Caco-2 cells cannot be grown as cysts, which hampers the visualization of the MT network.” 

Please take the time.

Concerning the Detyrosination of the MT network that  is responsible for cell shape changes. 

            Previous question was: Consequences of overexpressing VASH1/2 would add a plus 

Concerning the last point of reviewer 2. The effect of an overexpression of VASH 1/2 on detyr-tubulin levels are clearly an interesting issue. However, due to the limited time for the revision of the manuscript, it is not feasible to perform such experiments. Still, we will keep this in mind for our future work. 

Author Response

Again, we thank reviewer 2 for taking the time to read and evaluate our manuscript. Please find below our responses as a point-by-point reply.

The authors clarify a number of points in their answer. Once again, I find this is an interesting study.

I however insist that some time should be used to try to perform the very simple experiments, I asked for.

Regarding figure 5 now 4B: Please provide the original Data

“However, for the quantitation, all 3 experiments were evaluated, which is shown in revised figure 4b. If desired, we can provide the original data resulting from the densitometric analysis. “

The original data will be provided.

Please perform the experiments

            Previous question was: Why not use also the TTL antibody on western blot, that would strenghen the results.  “In the context of the revision of the manuscript we performed immunoblots regarding the TTL expression of Caco-2 cells after stimulation with FF, OVA and IFN-y. However, our primary investigations did not show clear effects regarding the TTL expression. For the detailed investigation of this issue, we would have to seed new Caco-2 cells and perform treatment of 7-14 day old cells with the mentioned substances again. Since the time period for the revision was limited, this was not feasible. However, if desired, we will perform additional experiments regarding this issue. “

Further experiments are ongoing and we expect the final results within January 2023.

I know that cells grown several days after confluency do not have a nice microtubule network.  The purpose of showing such micrograph is not to see a nice MT network, but eventually to see and analyze potential changes in its organization. I still think that high magnification of the MT network will be informative.

            Previous question was: In Figure 6, it would be nice to show a high magnification of the MT network (Total and Detyr tubulin) in control and IFN-g treated Caco-2 cells, to better visualize the changes in MT organization. 

- “We did not use high magnification images of Caco-2 cells since the visualization of the ultrastructural MT network was not satisfactory in the confluent polarized monolayers of Caco-2 cells. One reason for this is that unlike MDCK cells, Caco-2 cells cannot be grown as cysts, which hampers the visualization of the MT network.” 

Further experiments are planned and we expect the results within February 2023.

Please take the time.

Concerning the Detyrosination of the MT network that  is responsible for cell shape changes. 

            Previous question was: Consequences of overexpressing VASH1/2 would add a plus 

Concerning the last point of reviewer 2. The effect of an overexpression of VASH 1/2 on detyr-tubulin levels are clearly an interesting issue. However, due to the limited time for the revision of the manuscript, it is not feasible to perform such experiments. Still, we will keep this in mind for our future work. 

Even though we reckon this as an interesting aspect, the establishment of such an overexpression model of VASH 1/2 in Caco-2 cells is challenging and time consuming and its success is highly doubtful. Additionally, our manuscript deals with the tyrosination status in celiac disease and in a Caco-2 model of celiac disease. The expression of VASH 1/2 is not addressed in our article. Henceforth, we regard the demand of an overexpression model of VASH 1/2 as being beyond the scope of our manuscript. This point is the only one that delays and hinders the publication of our manuscript, which has been considered to be of interest for the readers of the “International Journal of Molecular Sciences”.

Round 3

Reviewer 2 Report

Ok for publication..